# Pace of passive margin tectonism revealed by U-Pb dating of fracture-filling calcite

William H. Amidon [1✉], Andrew R. C. Kylander-Clark[2], Matthew N. Barr[1], Samuel F. I. Graf[1] & David P. West Jr.[1]

A growing body of evidence demonstrates that Atlantic-style passive margins have experienced episodes of uplift and volcanism in response to changes in mantle circulation long after cessation of rifting. Passive margins are thus an attractive archive from which to retrieve records of mantle circulation and lithospheric alteration. However, this archive remains under-utilized due to difficulty in deciphering the surficial records of passive margin tectonism and linking them to seismic velocity structure. Here we present a new approach to unraveling the tectonic history of passive margins using U-Pb dating of calcite in faults and fractures along the eastern North American margin. These ages show a 40 Myr long period of continuous fracturing and faulting from ~115 to 75 Ma followed by another episode in Mio-Pliocene time. We argue that the former event represents a response to Cretaceous lithospheric alteration whereas the latter records development of modern relief in the northern Appalachians.

[1] Geology Department, Middlebury College, Middlebury, VT 05753, USA. [2] Earth Science Department, UC Santa Barbara, Santa Barbara, CA 93106, USA.
✉email: wamidon@middlebury.edu

Most Atlantic-type passive margins have experienced episodes of post-rift tectonic uplift, often accompanied by magmatism[1,2]. Although mantle processes are certainly involved, the exact mechanisms remain unclear and might be divided into three broad categories: 1) large-scale upwelling of the mantle such as plumes and hot-spots[3], 2) smaller scale edge convection at the margins of cratonic lithosphere[4], and 3) intraplate stresses arising from far-field plate reorganization[5]. Developing techniques to document and interpret the surficial records of these processes would allow passive margins to be used as sensitive recorders of past mantle processes. In particular, the spatio-temporal pattern of surficial tectonism may allow persistent processes like edge convection to be distinguished from transient or more localized processes like plate reorganization.

The rugged topography and Cretaceous igneous rocks of northern New England have long suggested post-rift lithospheric instability. Recent detailed seismic studies of mantle and lithospheric structure present a unique opportunity to link these features to near surface tectonism[6–9]. However, documenting recent surficial tectonism has proven challenging. For example, relatively small magnitudes of uplift and exhumation make it difficult to detect recent cooling events by thermochronology. Likewise, other signals of post-rift tectonism such as faulting or terrestrial sedimentation are poorly exposed or lacking, respectively. To overcome these challenges we use U-Pb dating of calcite in faults and fractures to develop a timeline of tectonic deformation in the Champlain Valley (CV) of Vermont and New York, directly above an area in which low-velocity seismic anomalies suggest significant lithospheric alteration (Fig. 1). This approach has the advantage of dating young brittle structures that bridge the temporal gap between relatively old thermochronologic ages (e.g. 75 Ma) and interpretation of young landforms. Although U-Pb dating of calcite by ID-TIMS has existed for many years[10], recent advancements in laser ablation inductively coupled mass spectrometry (LA-ICPMS) have dramatically improved sample throughput, creating a new opportunity to directly date brittle tectonic structures[11–14].

The CV represents the eastward edge of the North American cratonic lithosphere[15]. Thus, in the western part of the study area, the Adirondack (ADK) dome is primarily composed of Mesoproterozoic metamorphic rocks[16]. In contrast, the adjacent CV is composed of low grade meta-sedimentary rocks deposited on the passive margin of Laurentia in Cambro-Ordovician time[17]. Eastward of the CV, the Green Mountains of Vermont are composed of higher-grade metamorphic rocks emplaced during the Ordovician Taconic Orogeny. Importantly, the CV seems to have avoided deformation during subsequent Appalachian orogenic events and deformation associated with opening of the Atlantic Ocean basin in Late Triassic-Early Jurassic time.

Several lines of evidence suggest that the CV has experienced post-rift tectonic rejuvenation. First, widespread magmatic intrusions were emplaced from roughly 139 to 103 Ma as part of the larger New-England Quebec (NEQ) igneous province[18]. The trigger of NEQ magmatism is a long-standing debate. Some authors point to a regional west to east age progression as evidence for triggering by passage of the Great Meteor plume in the Early Cretaceous[19]. Others have argued for structurally controlled decompression melting[20], fluid driven melting[21,22], or a combination of multiple processes[23]. Second, northern New England is characterized by Late Cretaceous apatite fission-track (AFT) ages, younger than surrounding areas (Fig. S1 in Supp. Information). Finally, the physiography and relief of the CV suggests it is a graben structure that may have experienced Cenozoic tectonic deformation. For example, the western flank of the CV exhibits numerous topographic escarpments aligned with mapped extensional faults[24] and the ADK exhibit > 800 m of relief, steep slopes,

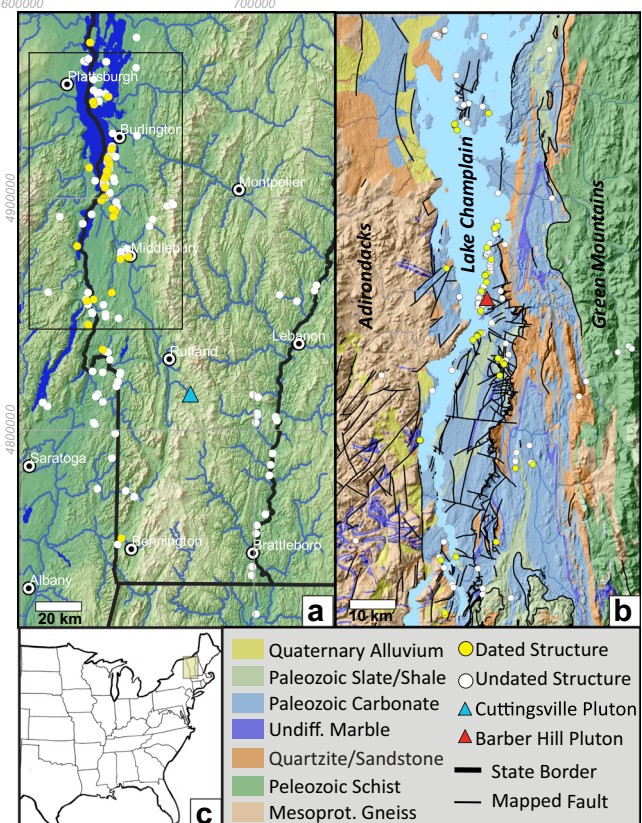

**Fig. 1 Maps of the study area. a** Physiographic map of the study area showing calcite samples. **b** Geologic map of the study area. The Champlain Valley represents the eastern edge of the North American Craton such that the western portion of the study area is composed of Proterozoic basement rocks and the eastern portion is composed of younger Paleozoic metamorphic rocks emplaced during the Taconic orogeny. The valley itself is composed of low grade Paleozoic meta-sedimentary rocks from which most samples were collected. Geologic maps from Ratcliffe et al., 2011 and Isachsen and Fisher, 1970. **c** Larger scale location map.

and a radial drainage network with active knick zones[25]. Miocene to ongoing differential uplift between the ADK and CV has been proposed using leveling lines[26] and fault analysis[27].

Although magmatism, Cretaceous cooling, and youthful relief have long been attributed to mantle-driven processes, recently documented seismic velocity anomalies provide specific geodynamic drivers. First, the CV sits at the edge of the North American craton where it abruptly thins from ~180 to 100 km beneath the western ADK near 75° longitude[15]. However, the fact that cratonic basement rocks are exposed eastward into Vermont suggests the cratonic lithosphere once extended further and its deeper portion was removed, perhaps during Cretaceous passage of the Great Meteor plume. Latest Cenozoic mantle bouyancy and resultant surface uplift is suggested by the Northern Appalachian Anomaly (NAA). The NAA is a 400 km wide low velocity zone beneath New England, centered at about 200 km depth in the upper asthenosphere with a sharp western boundary coincident with the eastern slope of the cratonic lithosphere. The NAA is currently experiencing vertical flow and is up to 700 °C warmer than surrounding asthenosphere, creating the potential for partial melt at the base of the previously thinned lithosphere[7,8,28–30]. Finally, very recent lithospheric alteration, and perhaps melting, is suggested by low velocity anomalies at shallow depths within the lithosphere beneath both the ADK and the southern Green Mountains. The ADK anomaly is 70–100 km wide at depths of

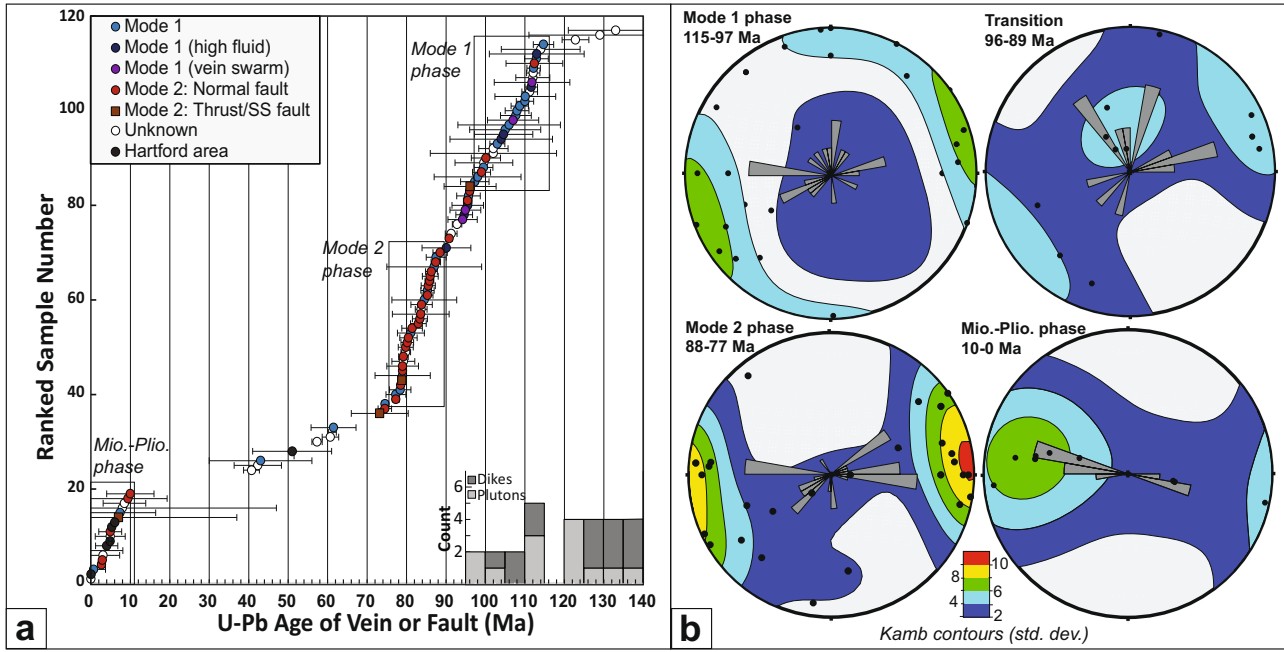

**Fig. 2 Chronology of fracturing and inferred σ₃ stresses. a** Chronology of dated calcite samples overlain with histogram of dated magmatic intrusions in the Champlain Valley. Data points represent the mean U-Pb isochron age and its associated 2σ error as computed by the Isoplot v4.15 AgeEr7Corr function. The onset of Cretaceous fracturing begins after a > 300 Myr tectonic hiatus and is divided into mode 1 and mode 2 phases based on the dominant style of fracturing observed during each time interval. Both phases of fracturing post-date initial magmatism and the mode 2 phase post-dates all of the magmatism in the Champlain Valley. The Miocene-Pliocene phase records a tectonic reawakening likely associated with the development of modern relief in northern New England. **b** σ₃ stresses inferred from field study of the dated faults and fractures. Note the relatively scattered stress directions for the Mode 1 phase compared with the more clustered E-W orientations for the Mode 2 phase.

50–85 km, most easily explained by the presence of partial melts[6]. The Green Mountain Anomaly is 100-150 km wide and observed at depths of 70–100 km beneath the southern Green Mountains[7]. The recent refinement of these low-velocity anomalies raises the question of exactly when lithospheric alteration occurred and how it was recorded in the surficial geology, which is the focus of this study. In this study we use the term lithospheric alteration to describe a broad range of potential processes that could cause low-velocity anomalies, including heating, geochemical alteration by fluids, or melting.

Here we present a new approach to unraveling the tectonic history of passive margins using U-Pb dating of calcite in faults and fractures in the Champlain Valley (CV). These ages show a 40 Myr long period of nearly continuous fracturing and faulting from ~115 to 75 Ma followed by a period of relative quiescence and then another episode in Mio-Pliocene time. We argue that the first event represents a response to Cretaceous lithospheric alteration whereas the latter records adjustments in response to positive buoyancy associated with development of the Northern Appalachian Anomaly and the development of modern relief in the northern Appalachians.

## Results

**U-Pb Dating of Calcite in Faults and Fractures**. U-Pb calcite ages from faults and fractures provide a timeline of tectonic deformation in the CV beginning with a cluster of ages from ca. 444-429 Ma centered on the transition from the Taconic to Salinic orogeny[37] (Supplementary Dataset 1). Structures of this age are found throughout the CV on N-S trending dilational veins or fault breccias (sites 7b, 18, 23, 43). Although these Taconic-Salinic ages are not discussed further, detailed site summaries for all sites can be found in source data files 1, 2.

The Taconic orogeny was apparently followed by a ~300 Myr long quiescent period that produced no dateable structures in the

CV. The was followed by a 40 Myr period of continuous fracture activity from ca. 115 to 75 Ma, which can be subdivided into two phases (Fig. 2a). First, a 'Mode 1 Phase' from roughly 115 to 97 Ma is dominated by sub-vertical mode 1 fractures (e.g. sites 1, 8, 11, 13, 14, 22, 25, 34, 40, 41) often associated with high fluid pressures (e.g. sites 2, 4, 18, 21B). Inferred σ₃ stresses during the Mode 1 Phase vary widely suggesting that stresses were either isotropic across the region or were time/space variable (Fig. 2b). A later 'Mode 2 Phase' from roughly 88 to 75 Ma is dominated by mode 2 fractures, typically brittle fault zones ranging from sub-meter to outcrop scale (e.g. sites 5, 6, 12–14, 19–21, 27, 33, 36). Inferred σ₃ stresses during this period are primarily E-W (Fig. 2b). Although some Mode 2 structures fall into the Mode 1 phase and vice versa, this is expected given the structural complexity of the study area.

The 'Mode 2 Phase' is followed by a second period of relative quiescence from ca. 75–10 Ma (Fig. 2a). This was followed by a 'Mio-Pliocene Phase' involving both Mode 1 and 2 structures that record high angle E-W extension, often along existing structures (e.g. sites 4, 6, 12, 31, 37). The widespread occurrence of a Mio-Pliocene fracturing phase is further supported by six Mio-Pliocene U-Pb ages on faults and fractures in the Hartford Graben (sites 28-30, 42). Although these structures are located outside of the CV, they also sit above the Northern Appalachian Anomaly and are included here as supporting evidence for a regional tectonic event in the Mio-Pliocene time.

**Stable isotope analyses of calcite**. Stable isotope results (Supplementary Dataset 2) suggest that calcite in faults and veins is derived from local source rocks because the observed range of δ¹³C and δ¹⁸O in veins is similar to the host rocks and heavier than typical values for mantle or meteoric fluids[38]. However, δ¹⁸O of calcite is typically lighter than paired host rock,

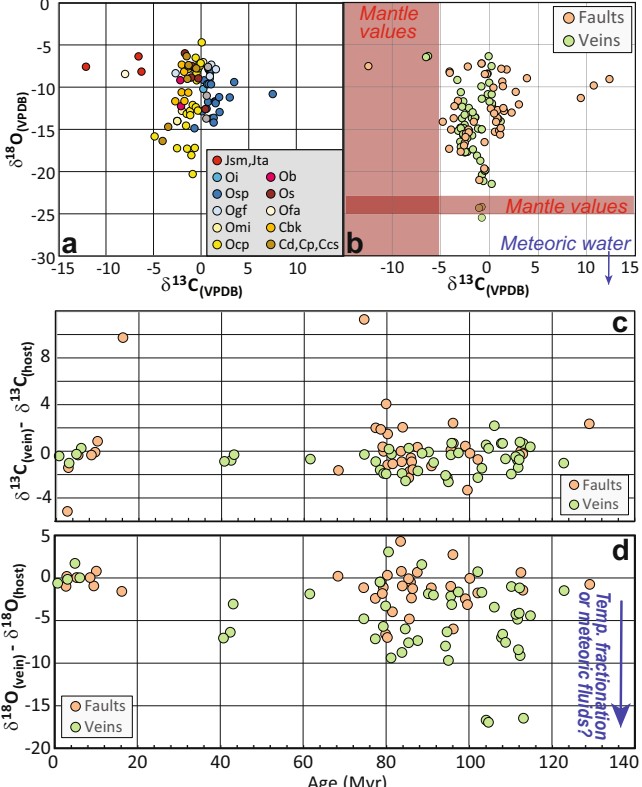

**Fig. 3 Stable isotopes measured in calcite and host veins. a** Stable isotope results for host rocks colored by rock type from Ratcliffe et al. 2011. Younger shale units (O$_{sp}$, O$_i$, and O$_{gf}$) show slightly heavier δ$^{13}$C values then the older Ordovician carbonates (O$_{mi}$, O$_{cp}$, O$_b$, O$_s$, O$_{fa}$, and O$_{bk}$). **b** Isotopic values of calcite from faults and veins, which appear to have crystallized from locally derived fluids because they do not overlap published values for mantle fluids or meteoric waters. **c, d** Difference between δ$^{13}$C and δ$^{18}$O measured in vein and host rock. Vein-host offsets in δ$^{18}$O are consistent with temperature fractionation between fluid and calcite. Particularly small offsets for Mio-Pliocene samples suggest calcite formed from locally derived fluids at a low temperature and has not been recrystallized.

suggesting temperature fractionation or a component of exotic fluids. Particularly small vein-host δ$^{18}$O offsets for Mio-Pliocene samples suggest this calcite was locally derived, crystallized at a relatively low temperature close to the surface, and has not been altered by meteoric fluids (Fig. 3).

**Methodological uncertainties.** One challenge of U-Pb dating of calcite is demonstrating that the U-Pb age of the calcite is representative of the timing of fracture formation or fault slip. To this end, samples are only collected from fresh unweathered exposures for which calcite crystallization can be clearly linked to fault slip or fracture formation. Nonetheless, roughly 10% of dated calcite samples have experienced multiple crystallization events. In the lab, secondary crystallization is identified based on the spatial pattern of high $^{238}$U/$^{206}$Pb laser shots combined with a visual inspection of the sample texture under an optical microscope. Samples that contain multiple isochron age domains and/ or multiple textural generations of calcite are deemed to have 'secondary crystallization' and their inferred stress is categorized as unknown. Ages of secondary crystallization are reported in supplementary dataset 1 and source data files 1, 2, but are not included in Fig. 2 or considered in the larger tectonic analysis.

Another uncertainty in U-Pb dating of calcite is assessing the accuracy of any given U-Pb age. Throughout the study we report 2σ analytical uncertainties on the isochron age as computed by Isoplot 4.15[35]. However, when the 2σ uncertainty on the isochron age is small (e.g. a tightly fit isochron line) it likely underestimates the true uncertainty due to undocumented analytical biases. For example, if the dated sample is not well matrix-matched to the primary standard then unknown systematic offsets in measured ratios can arise. Although this issue is thought be small based on good reproducibility of secondary calcite standards, each U-Pb age should nonetheless be treated as an estimate and tectonic interpretations should not rest entirely on a single site or sample. Another indicator of data consistency is the $^{207}$Pb/$^{206}$Pb ratio of the y-intercept on the isochron plot, typically interpreted as the initial Pb ratio. Our Y-intercept values range from 0.60 to 0.88, with a weighted average near 0.83. They show no correlation with the inferred isochron age (Fig. S2 in Supp. Information).

## Discussion

The prolonged 40 Myr period of fracturing from ca. 115 to 75 Ma likely records the upper crust adjusting to slow changes in asthenosphere-lithosphere dynamics rather than simply a short-lived pulse of tectonism. This is consistent with a period of prolonged volcanism in the NEQ province from ca. 139 to 98 Ma, including emplacement of at least 20 small plutons and countless dikes across southernmost Quebec, Vermont, New Hampshire and Southern Maine. Interestingly, the onset of mode 1 fracturing in the CV seems to have post-dated the period of peak NEQ magmatism, which occurred from roughly 128 to 118 Ma and included the Monteregian intrusions of Quebec, the Ascutney pluton in Vermont, and the Ossipee and Patuckaway intrusions of New Hampshire. However, Mode 1 fracturing temporally overlaps the emplacement of the Barber Hill and Cuttingsville plutons in the Champlain Valley, and the Oka intrusive complex near Montreal, which date to 112 ± 2 Ma, 98 ± 2 Ma, and 105 ± 6 Ma respectively[39,40]. Together with dikes spanning 139 to 106 Ma in age[41], these sites demonstrate the north-south axis of the Champlain Valley experienced the longest lived and youngest magmatism in the NEQ, the final 15 Myr of which was accompanied by Mode 1 fracturing. This suggests the CV is an axis of weakened lithosphere and/or a major structural boundary.

Additional hypotheses regarding the cause of faulting and fracturing arise from careful consideration of the structural style. First, the Mode 1 Phase was likely driven by fluid overpressure or cooling of the CV sedimentary rocks. Mode 1 fractures form under low differential effective stresses with a very low or tensile minimum confining stress. Such conditions are typically achieved by some combination of fluid overpressure, contraction during cooling, or tectonic stretching[42]. We favor the fluid overpressure or cooling hypotheses for several reasons. First, many sites show evidence for fluid overpressure (e.g. sites 2, 4, 18, 21B). Second, the Mode 1 phase post-dates peak NEQ plutonism from ca. 128-118 Ma, and thus could conceivably be driven by erosional unroofing of a volcanic plateau. Third, the wide range of inferred σ$_3$ stresses (Fig. 2b) are more consistent with isotropic contraction from cooling or locally variable fluid pressures than with tectonic stretching. Fourth, regional cooling during the mode 1 phase is indicated by 110-90 Ma AFT ages (Fig. S1 in Supp. Information).

The Mode 2 Phase from ca. 88–75 Ma post-dates magmatism by a least 10 Myr and seems to record predominantly E-W extension on brittle shear fractures, often with down-dip slip. Shear fractures form under higher differential stress often induced by tectonic stretching[43]. The timing overlaps a period of accelerated cooling and possible exhumation in the White Mountains of New Hampshire[44], a shift in the North

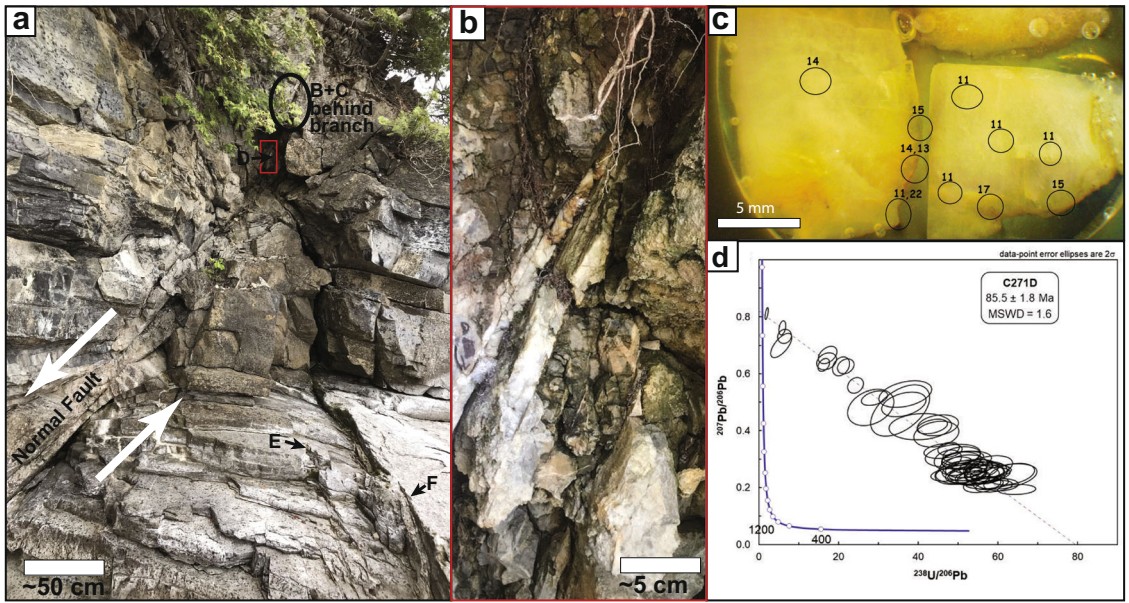

**Fig. 4 Example study site. a** Overview photo showing a top-to-the-east extensional fault zone at site 21a (Kingsland Bay State Park). The fault dips to the lower left corner of the photo (east) offsetting beds of Ordovician limestone. Six samples (A through F) were collected from this site, yielding ages from 79 to 88 Ma. Complete details and results from all sites can be found in source data files 1, 2. **b** Sample 271D was collected from tabular calcite within the fault zone. **c** Optical microscope photo of sample 271D annotated with the location of preliminary laser shots that showed high $^{238}$U/$^{206}$Pb ratios. This image was used to target attractive areas of the crystal during subsequent analysis at UC Santa Barbara. **d** Isotopic ratios measured at UC Santa Barbara. Each ellipse represents a single laser shot on the same calcite sample. They define a mixing line (isochron) between shots with a higher concentration of initial common-Pb (to left) and shots with a higher concentration of radiogenic lead (to right). The dashed line shows the unanchored isochron line fit to the data. Sample age is estimated based on its projected intersection with the thick blue concordia line, as computed by Isoplot v4.15.

America–Africa pole of rotation[5], and the onset of spreading between Labrador and Greenland[45]. This Late Cretaceous plate reorganization seems to have induced a shift in stress regime across the North Atlantic that caused faulting and fracturing throughout Scandinavia, Europe, and Africa[44].

The Mio-Pliocene Phase apparently records predominantly E-W $\sigma_3$ stress, typically recorded as minor fault slip within pre-existing high angle structures such as cleavage planes. One hypothesis is that distributed Mio-Pliocene strain records regional uplift and flexure in response to development of the Northern Appalachian, Green Mountain, or ADK anomalies, which are all considered extremely young[6,7]. Regional Mio-Pliocene uplift is supported by youthful topography, a pulse of Mio-Pliocene sedimentation in the Baltimore Canyon Trough[46], and five additional Mio-Pliocene ages from the Hartford Graben.

We conclude by proposing a speculative timeline for post-Jurassic tectonism in this portion of the northern Appalachian passive margin. First, arrival of the Great Meteor plume triggered kimberlitic and mafic intrusions in NY and Montreal area from roughly 150–136 Ma[47]. As North America passed over the plume, it triggered the main phase of NEQ plutonism from ca. 128–118 Ma, eroded the base of the lithosphere, and sent a thermal wave upward into the crust (e.g. [48]). As the hotspot moved offshore, the crust cooled and contracted, driving the Mode 1 fracturing phase from ca. 115–97 Ma and locking in the widespread Cretaceous AFT ages (Fig. S1 in Supp. Information). Isolated magmatism persisted in the CV until as late as 98 Ma, perhaps due to eastward migration of fluids upwards along the steep eastern edge of the cratonic lithosphere (e.g. Guimarães et al., 2020) or existence of deep-seated fractures. By the Late Cretaceous, plate reorganization changed regional stresses, causing E-W extension during the Mode 2 phase from ca. 88–75 Ma. After roughly 50 Myr of relative quiescence, the Northern Appalachian, Green Mountain, and ADK seismic anomalies developed in the Miocene driving regional uplift and recorded by the Mio-Pliocene phase of fracturing. We hypothesize that the Mio-Pliocene fracturing phase constrains the timing of recent lithospheric alteration beneath New England and provides an important argument in favor of tectonic rejuvenation in the long-running debate over the source of persistent geomorphic relief in the northern Appalachians[49].

## Methods

**U-Pb dating of calcite.** Samples were collected from roughly 600 veins and faults, primarily in the Champlain Valley (Fig. 1). Detailed U-Pb results and descriptions of all sites are available in individual site summaries found in source data files 1, 2. Because only ~13% of samples prove dateable, sample collection was semi-random, guided by the location of outcrop and existence of young structures that cross-cut bedding or Taconic-aged cleavage. Samples were collected from fresh, unweathered outcrops, mounted in 1" epoxy rounds, polished, and photographed prior to LA-ICPMS analysis (Fig. 4). Suitability for U-Pb dating was assessed by analyzing 50 to 100 random spots in the Middlebury College LA-ICPMS lab using a NWR 213 nm laser coupled to an iCAP-Q mass spectrometer. Samples with sufficiently high $^{238}$U/$^{206}$Pb ratios were then shipped to UC Santa Barbara for higher precision measurements using a Photon Machines Excite 193 nm laser coupled to a Nu Instruments Plasma MC-ICPMS.

Samples were analyzed by LA-ICPMS at UC Santa Barbara over a four-year period (Supplementary Dataset 1 and Source Data File 3). In a typical run calcite and NIST614 reference materials were analyzed every 10 analyses, and a two-stage reduction scheme was implemented. First, Iolite v.3.0[31] was used to correct the $^{207}$Pb/$^{206}$Pb for mass bias, detector efficiency, and instrumental drift, and to correct the $^{238}$U/$^{206}$Pb ratio for instrumental drift, using NIST614 as the primary reference material. 2 seconds were removed from both the beginning and end of all analyses, yielding a typical count time of 11 s. Next, the $^{238}$U/$^{206}$Pb ratio was corrected using a linear correction in Excel such that the primary calcite standard, WC-1, yielded 254 Ma on a Tera-Wasserburg (TW) diagram, anchored to a $^{207}$Pb/$^{206}$Pb value of 0.85[32]. Similar to the previous correction of the $^{207}$Pb/$^{206}$Pb ratio, this correction includes offset due to both mass bias and detector efficiency differences (i.e., there is no prior gain calibration for the Daly detector array). Accuracy and precision were monitored using a variety of secondary standards including ASH15[33] (2.96 Ma) and Duff Brown Tank[34] (64 Ma). Our full suite of U-Pb data, including reproducibility of primary and secondary standards is available in source data file 3. Full analytical methods are described in Kylander-Clark et al.[11].

U-Pb ages are then computed using an unanchored inverse isochron approach in Isoplot 4.15[35] (Fig. 2). Resultant inverse concordia plots of measured $^{238}U/^{206}Pb$ vs. $^{207}Pb/^{206}Pb$ define either: 1) a single-age isochron, 2) multiple linear isochron segments, or 3) no linear isochron segments (e.g. a cloud of laser shots). In case 2 the sample is considered to be composite, recording multiple episodes of crystallization. In this case multiple isochrons are visually identified and the isochron age is estimated from a subset of the laser shots. In case 3 the sample is considered uninterpretable. All isochron ages are unanchored and exclude laser shots with $^{238}U/^{206}Pb$ ratios <2 that do not fall on a clear trajectory with the higher $^{238}U/^{206}Pb$ shots in the isochron. The rationale is that these low-ratio shots could be derived from mineral inclusions that are high in common Pb or from calcite associated with an earlier crystallization event, thereby anchoring the isochron to an incorrect common Pb ratio.

After dating calcite from a given structure, efforts were made to interpret the stress conditions under which it formed. These faults and veins can be divided into two main groups: (1) Mode 1 fractures that are sub-vertical, planar, dilational, and lack obvious slip indicators, and (2) Mode 2 'in plane shear' fractures containing slip indicators such as calcite slickenfibres, tabs, or stepped tiling. A subset of the Mode 1 fractures show evidence for high fluid pressures, including implosion breccias and undulating vein swarms. Given the rare identification of conjugate structures we estimate $\sigma_3$ stresses using a set of simplified assumptions. For Mode 1 fractures, $\sigma_3$ is assumed perpendicular to the fracture plane. For Mode 2 fractures/faults with in-plane slip indicators, $\sigma_1$ and $\sigma_3$ are assumed to be co-planar in a plane orthogonal to the fracture, with $\sigma_1$ oriented at 70° to the slip vector and $\sigma_3$ oriented 20° from slip vector in opposite directions[36]. Finally, for Mode 2 fractures without slip indicators but with tabular calcite greater than 1 cm thick, we assume predominantly down-dip normal slip recognizing that extensional slip is most likely to create significant void space for precipitation of tabular calcite lenses.

**Stable isotope analysis of calcite.** $\delta^{13}C$ and $\delta^{18}O$ were determined on drilled powder from most dated samples and their paired host rock to assess the possibility of recrystallization and better understand the source of carbonate-saturated fluids (Supplementary Dataset 2). If multiple textures were visible in hand sample, multiple samples were collected from the same vein to test for secondary recrystallization. Roughly 0.1 gram of powder was drilled using a dremel tool and analyzed at the University of Florida or Union College stable isotope labs. At the University of Florida isotopic measurements were conducted using a Kiel III carbonate preparation device coupled with a Finnigan-Mat 252 isotope ratio mass spectrometer. Each batch of samples was measured with eight NBS-19 and two NBS-18 standards. All samples were dissolved in phosphoric acid at 70 °C and for 10 minutes. At Union College, carbonate powders were analyzed on a Thermo Delta V Advantage IRMS coupled to a GasBench II. Reference standards LSVEC, NBS-18, and NBS-19 were used for isotopic corrections and to assign the data to the appropriate isotopic scale using linear regression. Samples were placed in septa capped vials, flushed with He, and reacted with >100% phosphoric acid at 55 °C for at least 3 h.

## Data availability
All data generated during this study are included in this published article and its supplementary information files. All supplementary datasets and source data files are available in the Middlebury College data repository at https://doi.org/10.7926/71955140. Source data are provided with this paper.

## Code availability
No computer code was central to the conclusions of this manuscript, thus none is provided. Source data are provided with this paper.

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

## Acknowledgements
Thanks to many Middlebury students, Jody Smith, Keith Klepeis, Jon Kim, Greg Walsh, Jean Crespi, Jennifer Cooper-Boemmels, Jason Curtis, David Gillikin, Tim Schroeder, Emma Salazar, and Marley Amidon. This work was funded by award 1624170 from the NSF Tectonics program and by award UR-56956 from the Petroleum Research Fund.

## Author contributions
W.H.A. conceived of the idea, participated in field and lab work, and authored the manuscript. A.R.C.K. performed U-Pb analyses and primary data reduction. M.N.B., S.F.I.G., and D.P.W. performed field and lab work, data analysis, and interpretation.

## Competing interests
The authors declare no competing interests.
