## [Peer Review File · Nature Communications]

REVIEWER COMMENTS

Reviewer #1 (Remarks to the Author):

This paper concerns the post-rifting deformation history of eastern North America, which has been a lively topic for the last decade.

The major part of the paper concerns the ages of calcite deposited in fault and fractures, which is a proxy for the date of the faulting and fracturing. The authors make a compelling case that the ages are broadly distributed in the 0-10 Ma and 75-115 Ma periods, with few in the 10-75 Ma interval. This is an extremely important result that warrants publication of the paper, because it demonstrates that New England had been undergoing deformation during time periods when no volcanism is known to have occurred, and for broader time intervals than might be expected for a global plate-reorganization event or the passage of a hotspot. Consequently, it provides support for the notion that ongoing deep lithospheric or asthenospheric processes, such as delamination and/or edge convection, are responsible for the deformation.

A secondary part of the paper concerns new dates for igneous intrusions in the Lake Champlain region: a 104 Ma date for the Cuttingsville Pluton in the Burlington Lobe of the Champlain Valley igneous activity region, and a 130.6 Ma date for the Barber Hill Pluton in the Taconic Lobe. Both these dates are unexpected, but the very long 26.6 My time interval between them is remarkable, because while in different lobes, the two sites are separated by only 100 km (about the thickness of the lithosphere beneath them). This brings out a characteristic of New England volcanism that is radically different than other sites of continental volcanism, which are much shorter lived. For example, Yellowstone has been active only since about 2.1 Ma. At 15 Ma (the earliest date that I know), all the volcanism was at McDermitt caldera, in north-central Nevada, a thousand kilometers away.

A final part of the paper consists of an analysis of the 97 to 150 Ma volcanism in New England. In my opinion, this is the weakest part of the paper and should be omitted. My main criticism is that, because it is based on a more-or-less unvetted use of previously-published geochronologic data, its conclusions cannot be sufficiently well-founded to warrant publication in Nature Communications. The reason I think so is that the main use of the data is to claim that the volcanism "sweeps" west-to-east across New England. Recognition of this kind of pattern will be heavily biased by a few outliers (which are almost guaranteed in a compilation). Furthermore, the level of the data analysis is too qualitative, with the reader (e.g. in Figure 4) expected to discern a spatial pattern that is not well-articulated. I do agree that the Ontario-NY kimberlites are (mostly) older than the New England volcanism, and that the New England Seamounts are (mostly) younger. But it seems to me that the word "sweeping" implies a finer-scale and better-organized age progression, and I do not see such a progression in Figure 4.

In summary, the calcite age part of the paper is compelling and important. The "sweeping pattern" is not compelling, but could be deleted from the paper without affecting the calcite age part. Hence, I suggest that the paper be returned to the author for a shortening of this sort.

...

As an aside, I note that the authors say (e.g. on line 299) that the 0-10 Ma faults/fractures imply a stress regime with east-west extension. This is in contrast to present day earthquakes, "with near-horizontal P axes striking northeasterly" (ref below). Why should they be different?

Reference

Earthquake Source Parameters and State of Stress for the Northeastern United States and Southeastern Canada from Analysis of Regional Seismograms, Wen-xuan Du; Won-Young Kim;

Lynn R. Sykes, *Bulletin of the Seismological Society of America* (2003) 93 (4): 1633–1648.

Reviewer #2 (Remarks to the Author):

This is a thorough geochronological and isotopic study of the history of igneous and structural events in a region near the Atlantic margin of North America. I am impressed by the quantity and quality of data and I think that the work will be of great interest to a variety of specialists interested in the tectonics of continental margins as well as to those dating structural and fluid movement events. I have made comments below and in an annotated copy of the manuscript that I hope will improve the manuscript.

I find the author's use the term 'alteration' a bit confusing. I think of alteration as a low temperature chemical change in minerals and rocks but the authors seem to use it to signify any thermal overprint.

The manuscript needs some reorganization. Methods (section 2) presents results, especially for non-calcite samples. There is no section 3 and section 4 (Results) contains a lot of discussion of data.

The authors need to graphically show inverse concordia results for each sample in Appendix3. This could be done using chart sheets in the file. The information in the age columns in the Appendix3 Tables is linked to other files in directories that exist only on the author's computer. All information in the tables should be in the form of text.

In Table S1, which summarizes the ages, it would be good to provide a reference to the file in the Appendix that contains the data. Linking the ages to the concordia diagrams, if possible, would provide even easier access.

The authors make no mention of the common Pb parameter. The results of data regression yield estimates for age and the $^{207}\text{Pb}/^{206}\text{Pb}$ ratio of the initial common Pb component. They should both be given in Table S1. The common Pb isotopic ratio is usually better determined than the age for U-Pb data sets on calcite, which are often dominated by common Pb. Although the $^{207}\text{Pb}/^{206}\text{Pb}$ ratio is often uniform, reflecting the average composition of the crust through which the fluids passed, in some cases anomalously low values can distinguish a hydrothermal from a normal groundwater source.

In Fig 2 (and lines 240-249), it isn't clear to me how the authors decided on where to delineate Mode 1 and Mode 2 phases. The choice of colours for the symbols makes it somewhat difficult to distinguish samples. Mode 2 samples are not distinguished in the figure but I can see Mode 1 samples that are much younger than the Mode 1 field. Perhaps the age distributions of the two modes could be better shown using the probability density distribution function in Isoplot.

Don Davis

Response to reviewer comments (in blue)

Reviewer #1 (Remarks to the Author):

This paper concerns the post-rifting deformation history of eastern North America, which has been a lively topic for the last decade. The major part of the paper concerns the ages of calcite deposited in fault and fractures, which is a proxy for the date of the faulting and fracturing. The authors make a compelling case that the ages are broadly distributed in the 0-10 Ma and 75-115 Ma periods, with few in the 10-75 Ma interval. This is an extremely important result that warrants publication of the paper, because it demonstrates that New England had been undergoing deformation during time periods when no volcanism is known to have occurred, and for broader time intervals than might be expected for a global plate-reorganization event or the passage of a hotspot. Consequently, it provides support for the notion that ongoing deep lithospheric or asthenospheric processes, such as delamination and/or edge convection, are responsible for the deformation.

A secondary part of the paper concerns new dates for igneous intrusions in the Lake Champlain region: a 104 Ma date for the Cuttingsville Pluton in the Burlington Lobe of the Champlain Valley igneous activity region, and a 130.6 Ma date for the Barber Hill Pluton in the Taconic Lobe. Both these dates are unexpected, but the very long 26.6 My time interval between them is remarkable, because while in different lobes, the two sites are separated by only 100 km (about the thickness of the lithosphere beneath them). This brings out a characteristic of New England volcanism that is radically different than other sites of continental volcanism, which are much shorter lived. For example, Yellowstone has been active only since about 2.1 Ma. At 15 Ma (the earliest date that I know), all the volcanism was at McDermitt caldera, in north-central Nevada, a thousand kilometers away.

Thanks for this comment, it is very interesting to consider why the Champlain Valley experiences such long-lived volcanism and is definitely a point that I would like readers to take away from this paper. To clarify, the original manuscript did present new dates on the Cuttingsville Pluton, but they simply confirm what has long been known about the age of the youngest volcanism in the Champlain Valley, which is well summarized by Boemmels et al., 2021. K/Ar ages of roughly 100 Ma in the Cuttingsville were first published by Armstrong and Stump, 1971 and more recently confirmed using U/Pb dating of zircon by Boemmels et al., 2021. The older ages for magmatism in the Champlain Valley (~139 Ma) have also been reported by multiple sources over the years. Our new ages are in excellent agreement, but we have removed them from the paper as described below.

A final part of the paper consists of an analysis of the 97 to 150 Ma volcanism in New England. In my opinion, this is the weakest part of the paper and should be omitted. My main criticism is that, because it is based on a more-or-less unvetted use of previously-published geochronologic data, its conclusions cannot be sufficiently well-founded to warrant publication in Nature Communications. The reason I think so is that the main use of the data is to claim that the that volcanism “sweeps” west-to-east across New England. Recognition of this kind of pattern will be heavily biased by a few outliers (which are almost guaranteed in a compilation). Furthermore, the level of the data analysis is too qualitative, with the reader (e.g. in Figure 4)

expected to discern a spatial pattern that is not well-articulated. I do agree that the Ontario-NY kimberlites are (mostly) older than the New England volcanism, and that the New England Seamounts are (mostly) younger. But it seems to me that the word “sweeping” implies a finer-scale and better-organized age progression, and I do not see such a progression in Figure 4. In summary, the calcite age part of the paper is compelling and important. The “sweeping pattern” is not compelling, but could be deleted from the paper without affecting the calcite age part. Hence, I suggest that the paper be returned to the author for a shortening of this sort.

We disagree that the sweeping pattern is speculative, given that it appears clearly in the various panels of figure 4. Likewise, we disagree that the compilation of radiometric ages is not well vetted. We would welcome feedback on specific ages the reviewer feels should be dropped from the compilation. One possible criticism is that we are willing to include K/Ar ages on whole rock where perhaps others throw these out. That said, those ages are relatively few and do not change the core conclusions of the analysis.

Nonetheless, we agree that combining the calcite and the magmatic ages in the paper is too much and are happy to remove the magmatic ages and our age compilation from the paper. We have thus removed figure 4, tables S3 to S6, and any portions of the text referring to those data. Our discussion of magmatic ages is now focused in the beginning of the discussion section and is based primarily on the recent work by Boemmels et al., 2021 and Kinney et al., 2021. These two papers support our basic conclusion that magmatism was very long-lived in the Champlain Valley and the youngest volcanism post-dates the peak of magmatism by about 20 Myr.

...

As an aside, I note that the authors say (e.g. on line 299) that the 0-10 Ma faults/fractures imply a stress regime with east-west extension. This is in contrast to present day earthquakes, “with near-horizontal P axes striking northeasterly” (ref below). Why should they be different?

Reference

Earthquake Source Parameters and State of Stress for the Northeastern United States and Southeastern Canada from Analysis of Regional Seismograms, Wen-xuan Du; Won-Young Kim; Lynn R. Sykes, Bulletin of the Seismological Society of America (2003) 93 (4): 1633–1648.

This is a very interesting point, thank you. Our current thinking is that the Mio-Pliocene phase in the Champlain Valley records a regional uplift event in response to positive buoyancy associated with development of the North American Anomaly. This may have caused localized extensional faulting as the lithosphere flexed outward. Alternatively, E-W σ_3 stresses may record strain localization within the Champlain Valley where most structures trend N-S and two blocks of metamorphic basement flank a down-dropped sedimentary platform. For example, such structures could form as the Adirondack and Green Mountain blocks moved upwards relative to the Champlain Valley along N-S trending extensional faults.

Finally, it is worth noting that the accuracy of the Mio-Pliocene U-Pb ages is so poor that they may not represent modern times despite a few that have estimated ages <1 Ma. For example, in many cases we can only say that the age is < 15 Ma, although many appear to be as young as 5 Ma. Regardless, it may not be reasonable to expect the same state of stress exists today as did 5 Myr ago.

Reviewer #2 (Remarks to the Author):

This is a thorough geochronological and isotopic study of the history of igneous and structural events in a region near the Atlantic margin of North America. I am impressed by the quantity and quality of data and I think that the work will be of great interest to a variety of specialists interested in the tectonics of continental margins as well as to those dating structural and fluid movement events. I have made comments below and in an annotated copy of the manuscript that I hope will improve the manuscript.

I find the author's use of the term 'alteration' a bit confusing. I think of alteration as a low temperature chemical change in minerals and rocks but the authors seem to use it to signify any thermal overprint.

Thanks for this helpful comment. We have added text at line 88 to clarify what we mean:

"In this study we use alteration to describe a broad range of potential processes that could cause low-velocity anomalies, including heating, geochemical alteration by fluids, or melting."

We have also clarified this near line 42 of the revised manuscript with text that now reads:

"...directly above an area in which low-velocity seismic anomalies suggest significant lithospheric alteration."

The manuscript needs some reorganization. Methods (section 2) presents results, especially for non-calcite samples. There is no section 3 and section 4 (Results) contains a lot of discussion of data.

We are very appreciative of this comment and acknowledge that there are definitely some generalized results presented in the methods section. We have carefully considered which parts of the methods section we might move into the results and would respectfully prefer to leave the methods section intact as written. We have moved the last two paragraphs of the methods section into the later part of the results section near lines 196-213.

We have not moved other parts of the methods section into the results. Our reasoning is that it is necessary to present some generalized results in order to justify certain methods. For example, we need to explain that only 13% of calcite veins are dateable in order to justify semi-random sampling. Or we need to explain that some samples display scattered U/Pb ratios to justify our approach to isochron fitting. Likewise, we need to give a basic explanation of the types of structures we examined in order to explain the methods we used to infer stresses. We do wait until the results section to discuss specific results like U-Pb ages and stress orientations. Although we feel our current organization yields a higher

readability, we are willing to rewrite the methods section if the reviewer feels strongly. Thanks for your consideration.

The authors need to graphically show inverse concordia results for each sample in Appendix 3. This could be done using chart sheets in the file. The information in the age columns in the Appendix 3 Tables is linked to other files in directories that exist only on the author's computer. All information in the tables should be in the form of text.

Concordia diagrams for all dated samples can be found towards the bottom of the site summaries available in Appendix 1 (formerly appendix 2). The relevant site numbers can be found in column A of Table S1 or directly mentioned in the text. That said, we have now modified the spreadsheets in Appendix 2 (formerly appendix 3) so that they also include concordia diagrams for dated samples.

We have also removed any Isoplot formulas from the Excel files so there are no longer broken links upon opening the files. Thanks for pointing that out.

In Table S1, which summarizes the ages, it would be good to provide a reference to the file in the Appendix that contains the data. Linking the ages to the concordia diagrams, if possible, would provide even easier access.

This is a very helpful comment. The current idea is that the site number in column A of table S1 can be used to reference the site summary in appendix 1 (formerly appendix 2), which contain the concordia diagrams as well as field and laboratory photos of the sample. That said, we agree more could be done to guide readers to the relevant data! We have now added a 'raw data sheet' column to table S1, which contains the name of the relevant spreadsheet containing data and concordia plots. We have also added text to the top of table S1 directing readers to the site summary files in appendix 1.

The authors make no mention of the common Pb parameter. The results of data regression yield estimates for age and the $^{207}\text{Pb}/^{206}\text{Pb}$ ratio of the initial common Pb component. They should both be given in Table S1. The common Pb isotopic ratio is usually better determined than the age for U-Pb data sets on calcite, which are often dominated by common Pb. Although the $^{207}\text{Pb}/^{206}\text{Pb}$ ratio is often uniform, reflecting the average composition of the crust through which the fluids passed, in some cases anomalously low values can distinguish a hydrothermal from a normal groundwater source.

We use unanchored regression when fitting the isochrons on inverse Concordia plots. This method avoids the need to assume a common-Pb parameter (ratio). Here is the text (retained from the original manuscript) that describes our approach: “All isochron ages are unanchored and exclude laser shots with $^{238}\text{U}/^{206}\text{Pb}$ ratios < 2 that do not fall on a clear trajectory with the higher $^{238}\text{U}/^{206}\text{Pb}$ shots in the isochron. The rationale is that these low-ratio shots could be derived from mineral inclusions that are high in common Pb or from calcite associated with an earlier crystallization event, thereby anchoring the isochron to an incorrect common Pb ratio.”

In Fig 2 (and lines 240-249), it isn't clear to me how the authors decided on where to delineate Mode 1 and Mode 2 phases. The choice of colours for the symbols makes it somewhat difficult to distinguish samples. Mode 2 samples are not distinguished in the figure but I can see Mode 1 samples that are much younger than the Mode 1 field. Perhaps the age distributions of the two modes could be better shown using the probability density distribution function in Isoplot.

Thanks for this helpful comment! We have now modified the legend to clearly label which symbols fall into the “mode 2” category, which is a huge clarity improvement! There are definitely Mode 1 structures that fall into the Mode 2 phase and vice versa, but we argue this is expected from complex natural systems. We have added text to the paper near line 177 justifying this: “Although Mode 2 structures fall into the Mode 1 phase and vice versa, this is expected given the structural complexity of the study area”.

We experimented with a probability density diagram but did not find an aesthetic that we liked. One challenge with that approach is that probability density contributed from a given sample corresponds to the size of the analytical error on that sample. This creates huge spikes for samples whose ages happen to be very precise. This is misleading because the precision of those ages is better than the accuracy. We thus prefer the more ‘kernelized’ approach that we used.

Regarding color, the contrast between blue and red seems adequate, hopefully improved labeling in the legend has helped with this issue. Please let us know if not.

REVIEWER COMMENTS

Reviewer #1 (Remarks to the Author):

This is a much cleaner and well-focused version than the original. The major result, two stages of tectonic activity with a long hiatus between them, is well-substantiated and important, and will be of interest to a diverse group of geodynamicists, geophysicists and geochemists studying the New England area in general and (not so) passive margins in general. I recommend that the paper be published.

Reviewer #2 (Remarks to the Author):

This work presents an enormous amount of data that is mostly very well documented. The age results are given in a general table like '337863_1_data_set_6151529_r436r3.xlsx' that includes structural information. This is presumably Table S1 in the manuscript, but does not seem to be labeled as such, which caused me some confusion. The authors should make sure that files referred to in the text have the same names in the Appendix.

Aside from that, I find the revision to Amidon et al. to be acceptable except for the absence of a table presenting regression details. This should include best-fit ages and errors, initial common $^{207}\text{Pb}/^{206}\text{Pb}$ ratios (Y-intercept) and errors, number of data and MSWD for each isochron. I don't think that the authors understood the purpose of my previous comment below on initial common Pb compositions. Regression yields both age and Y-intercept estimations. While the Y-intercept is usually close to average crustal Pb isotopic compositions it need not be and can provide interesting information on the source of the fluids. Documenting it is, in any case, necessary to completely describe the regression. If there is a significant variation, it might be interesting to plot age versus initial $^{207}\text{Pb}/^{206}\text{Pb}$ ratios as was done in Dix et al. *Chemical Geology* 586 (2021) 120582. doi.org/10.1016/j.chemgeo.2021.120582

Previous reviewer comment:

The authors make no mention of the common Pb parameter. The results of data regression yield estimates for age and the $^{207}\text{Pb}/^{206}\text{Pb}$ ratio of the initial common Pb component. They should both be given in Table S1. The common Pb isotopic ratio is usually better determined than the age for U-Pb data sets on calcite, which are often dominated by common Pb. Although the $^{207}\text{Pb}/^{206}\text{Pb}$ ratio is often uniform, reflecting the average composition of the crust through which the fluids passed, in some cases anomalously low values can distinguish a hydrothermal from a normal groundwater source.

Author response:

We use unanchored regression when fitting the isochrons on inverse Concordia plots. This method avoids the need to assume a common-Pb parameter (ratio). Here is the text (retained from the original manuscript) that describes our approach: "All isochron ages are unanchored and exclude laser shots with $^{238}\text{U}/^{206}\text{Pb}$ ratios < 2 that do not fall on a clear trajectory with the higher $^{238}\text{U}/^{206}\text{Pb}$ shots in the isochron. The rationale is that these low-ratio shots could be derived from mineral inclusions that are high in common Pb or from calcite associated with an earlier crystallization event, thereby anchoring the isochron to an incorrect common Pb ratio."

Don Davis

Response to Reviewer Comments:

Reviewer #2 (Remarks to the Author):

This work presents an enormous amount of data that is mostly very well documented. The age results are given in a general table like '337863_1_data_set_6151529_r436r3.xlsx' that includes structural information. This is presumably Table S1 in the manuscript, but does not seem to be labeled as such, which caused me some confusion. The authors should make sure that files referred to in the text have the same names in the Appendix.

Thanks for raising this concern. We have confirmed that table S1 is correctly labeled and appendices are correctly referenced.

Aside from that, I find the revision to Amidon et al. to be acceptable except for the absence of a table presenting regression details. This should include best-fit ages and errors, initial common $^{207}\text{Pb}/^{206}\text{Pb}$ ratios (Y-intercept) and errors, number of data and MSWD for each isochron. I don't think that the authors understood the purpose of my previous comment below on initial common Pb compositions. Regression yields both age and Y-intercept estimations. While the Y-intercept is usually close to average crustal Pb isotopic compositions it need not be and can provide interesting information on the source of the fluids. Documenting it is, in any case, necessary to completely describe the regression. If there is a significant variation, it might be interesting to plot age versus initial $^{207}\text{Pb}/^{206}\text{Pb}$ ratios as was done in Dix et al. *Chemical Geology* 586 (2021) 120582. doi.org/10.1016/j.chemgeo.2021.120582

Thanks for clarifying this. Indeed, I completely missed the point the reviewer was making. We have now added columns to table S1 to include the number of points included in the regression, MSWD, initial Pb value, and 2s uncertainty. We have also plotted the distribution of initial Pb ratios in a new figure S2 to demonstrate a weighted mean value near 0.83 with only a few significant outliers. We have also plotted initial Pb ratio vs. U-Pb age to demonstrate a lack of correlation. This implies that the lead isotopes have not been overprinted and supports our interpretation of the isochron ages as primary crystallization ages.

I have added supporting text near lines 213-216 that reads:

Another indicator of data consistency is the $^{207}\text{Pb}/^{206}\text{Pb}$ ratio of the y-intercept on the isochron plot, typically interpreted as the initial Pb ratio. Our Y-intercept values range from 0.60 to 0.88, with a weighted average near 0.83. They show no correlation with the inferred isochron age (Fig. S2b).

The caption for the new supplementary figure reads:

S2) Summary of $^{207}\text{Pb}/^{206}\text{Pb}$ common lead ratios. a, Probability density plot (PDP) and histogram showing the distribution of y-intercept values for each calcite U-Pb age. These values are obtained by fitting an isochron line in $^{238}\text{U}/^{206}\text{Pb}$ vs. $^{207}\text{Pb}/^{206}\text{Pb}$ space and taking the upper Y-intercept. They are interpreted as the common lead ratio present in the crystal prior to radiogenic Pb ingrowth. b, Plot of calcite U-Pb age vs. y-intercept values. The lack of correlation implies isochron ages are not influenced by the common lead ratios and that most isochron ages record primary crystallization.

This exercise has GREATLY improved the presentation of the data and I am very grateful to the reviewer for bringing it up. I am hopeful this will help readers more quickly evaluate the quality of our data without needing to open every detailed site summary file. THANK YOU for the suggestion. Also, we have now cited the Dix et al., 2021 paper.

REVIEWERS' COMMENTS

Reviewer #2 (Remarks to the Author):

I am satisfied that the authors have addressed all of my previous concerns in their second revision and that the Amidon et al. manuscript is now acceptable for publication.

Don Davis